# Description of a Sarcoptic Mange Outbreak in Alpine Chamois Using an Enhanced Surveillance Approach

**DOI:** 10.3390/ani12162077

**Published:** 2022-08-15

**Authors:** Federica Obber, Roberto Celva, Martina Libanora, Graziana Da Rold, Debora Dellamaria, Piergiovanni Partel, Enrico Ferraro, Maria Santa Calabrese, Lia Morpurgo, Simone Roberto Rolando Pisano, Carlo Vittorio Citterio, Rudi Cassini

**Affiliations:** 1Istituto Zooprofilattico Sperimentale delle Venezie, 35020 Legnaro, PD, Italy; 2Paneveggio-Pale di San Martino Natural Park, 38054 Tonadico, TN, Italy; 3Associazione Cacciatori Trentini, 38121 Trento, Italy; 4Wildlife Management Office Province of Trento, 38121 Trento, Italy; 5Department of Animal Medicine, Production and Health (MAPS), University of Padua, 35020 Legnaro, PD, Italy; 6Centre for Fish and Wildlife Health (FIWI), Department of Infectious Diseases and Pathobiology, Vetsuisse Faculty, University of Bern, 3012 Bern, Switzerland

**Keywords:** surveillance, chamois, *Sarcoptes scabiei*, caprinae, density, mortality, sarcoptic mange

## Abstract

**Simple Summary:**

Sarcoptic mange represents an important concern for chamois management; in our study, the effects of an epidemic were monitored on an alpine population from 2006 to 2020. Passive surveillance and demographic data were analyzed in order to describe a mange outbreak. Furthermore, an enhanced passive surveillance protocol was implemented in a subpart of the study area in order to evaluate the efficiency of the ordinary one. Generally, the demographic decline caused by the epidemic reached the highest values between the first and the third year after the first mange cases. The enhanced passive surveillance approach proved to be an important asset for disease surveillance: However, its adoption may be too costly if applied for longer periods on a wide scale. Passive surveillance, in both ordinary and enhanced surveillance protocol, should encompass the use of other monitoring strategies in the future to study the eco-epidemiology of this disease in wild Caprinae.

**Abstract:**

Since 1995, the Alpine chamois (*Rupicapra r. rupicapra*) population of the Dolomites has been affected by sarcoptic mange with considerable management concerns. In this study, 15 years (2006–2020) of passive surveillance and demographic data were analyzed in order to describe a mange outbreak. Furthermore, an enhanced passive surveillance protocol was implemented in order to evaluate the efficiency of ordinary vs. enhanced surveillance protocol in identifying dead chamois in the field and in reaching a correct diagnosis. Our results confirm the role of mange as a determining factor for chamois mortality, while stressing the importance of a wider view on the factors affecting population dynamics. The enhanced passive surveillance protocol increased the probability of carcass retrieval and identification of the cause of death; however, its adoption may be too costly if applied for long periods on a wide scale. Passive surveillance, in both ordinary and enhanced surveillance protocol, should encompass the use of other strategies in the future to study the eco-epidemiology of the disease in wild Caprinae.

## 1. Introduction

Sarcoptic mange is a common and widespread disease caused by the burrowing mite *Sarcoptes scabiei*, which is arguably the most important mite species in terms of the number of domestic and wildlife species affected [1]. It represents a threat for wildlife, given its potential severity and morbidity, the wide range of host species affected and its global distribution [1,2].

It is probably the most severe disease affecting wild Caprinae in Europe [3,4], as testified by the mange epizootics that affected the Spanish ibex (*Capra pyrenaica*) populations in southeastern Spain [5,6] and the Alpine chamois populations in Austria [7] and north-eastern Italy [8]. Sarcoptic mange can have serious consequences, particularly for endangered species in which the loss of even a few individuals can be critical for its conservation [9]. This infection can be particularly threatening for small-sized, fragmented populations, as testified by the cases in Spanish ibex [5,6] and Alpine ibex [10]. In game species, the significant population crashes induced by mange is critical for wildlife management, forcing stakeholders to adapt hunting-bag sizes to cope with less abundant stocks [4,8].

Mortality by sarcoptic mange typically peaks in winter and spring, and it interacts with natural factors such as winter starvation and other climate constraints [8]. Life history of affected populations, namely previous contacts (or not) with the agent, also play an important role, as a first epidemic wave is likely to have remarkable effects on naïve populations. In the recently affected Dolomite Alps and Cantabrian Mountains, the chamois population size decreased on average by approximately two-thirds [8,11] and up to more than 80%. A 98% decrease rate occurred in a naïve and particularly sensitive population of Spanish ibex in Southern Spain [12]. Alternatively, in the case of successive contacts (usually occurring in form of minor waves at 10–15 years intervals), mortality rarely exceeds a value of 25% [13].

Since 1995, the Alpine chamois and ibex populations of the Dolomites area (North-Eastern Italy, across Belluno, Trento and Bolzano provinces) have been affected by mange originating from bordering Austria, entering Italy through the territory of the Belluno province of the Veneto Region. Consequently, local wildlife managers and veterinarians improved the coordination of their interventions for a better monitoring of mange cases. The epidemic front has progressively spread along a northeast towards southwest axis, with a raw average speed of 4.64 ± 3.12 km/year [14]. The oil-spot-like spread of the mange cases was observed, as well as apparent “jumps” followed by the filling of the rear gaps [8,15].

In the Trento Province, a specific local mange strategy was promulgated in 2001 aimed at monitoring the spread of the infection for an informed hunting management of the chamois population (Appendix A). 

The present study summarises the results of 15 years (2006–2020) of passive surveillance and demographic data, collected on the basis of the above-mentioned mange strategy. In addition, the study aimed at evaluating the efficiency of an enhanced passive surveillance performed in a small area found inside the study area.

## 2. Materials and Methods

### 2.1. Study Area

The study area is located in the northeastern part of the Trento province, and it comprises the entire territory of the Paneveggio-Pale di San Martino Nature Park (PPSM); it covers approximately 727 km^2^ (Xmin 46°5′41.94″, Xmax 46°22′42.65″; Ymin 11°22′12.48″, Ymax 11°57′47.63″), and it is entirely embedded in subsection 1A2b of Italian Ecoregions (Dolomiti and Carnia). Altitude ranges approximately from 560 to 3192 m.a.s.l., with a median altitude at 1700 m.a.s.l. The local orography is strongly influenced by Pale di San Martino dolomitic massif, as well as the peaks of the Lagorai range.

The local fauna includes several wild ruminant species, such as Alpine chamois (*Rupicapra rupicapra* L., 1758), Alpine ibex (*Capra ibex* L., 1758), red deer (*Cervus elaphus* L., 1758), roe deer (*Capreolus capreolus* L., 1758) and mouflon (*Ovis musimon* P., 1762).

Chamois is a renowned game species overall in the Trento province; for hunting purposes, populations are monitored at local level using block count; the data are then aggregated by larger areas, inside which the overall hunting pressure must not exceed 15% of the census population and usually settles at about 10%. Management of chamois population underwent significant changes in the study area, since the detection of the first mange cases in 2005: The study area has been further subdivided into 12 “mange zones”, identified by geographical criteria (e.g., presence of deep valleys, large rivers and artificial barriers) and, where possible, by chamois population units, in a view to implement more targeted preventive and reactive measures. These included the following:Mange management strategy in chamois populations with applications of seven types of hunting strategies (including hunting suspension), inside the mange zones, according to the epidemiological situation as described in Appendix A;Definition of mange case: cadavers with typical skin lesions confirmed by a laboratory diagnosis of *S. scabiei*.A general systematic passive-surveillance (hereafter termed “ordinary surveillance”) protocol in order to estimate the impact of mange on chamois populations (details are provided in Section 2.3);Training of game wardens and local hunters for the detections of cases and the definition of protocols for the proper handling of mangy chamois;Deployment of standardized population censuses using the methodology of block count in order to estimate the demographic decrease (details are provided in Section 2.2).

During 2008–2010, a small area (Figure 1), named Cavallazza Area (CA hereafter), was dedicated to the deployment of a more intensive protocol for census and of an enhanced passive surveillance (hereafter “enhanced surveillance”). The CA covers approximately 10 km^2^ (Xmin 46°15′59.01″, Xmax 46°17′55.49″; Ymin 11°45′39.21″, Ymax 11°48′13.49″) at a median altitude of 1938 m.a.s.l., ranging from 1563 to 2323 m.a.s.l.; it is entirely included in PPSM Nature park. Mange zones are shown in Figure 1.

### 2.2. Chamois Demographic Data

Chamois’ demographic status was investigated using block count data collected from 2006 to 2020. Block count is arguably the most widely used technique to monitor chamois populations in high-altitude open areas [16,17] and consists of counting animals from fixed vantage points or along trails using optical instruments [17]. The chamois survey area was subdivided into smaller “blocks” of 50 to 250 ha: that is, sectors chosen on the basis of natural boundaries such as valleys, ridges and rivers to favor animal detectability. A team of two to four operators (agents of the Trento Forestry Department, gamekeepers, PPSM park wardens or hunters) was assigned a fixed itinerary within each “block” area. During counts, operators noted, for each group of chamois, the number of animals, their sex and age class. The chamois was observed by means of 7–10× binoculars and 20–60× telescopes, and classified into six classes: kids (<1 year), yearlings (<2 years), adult males, adult females, adult undetermined and undetermined. Putative double counts were deleted by the census manager in the following 24 h. The highest animal count was assumed as the “minimum certain number” of chamois in that particular year.

Population density was calculated as chamois/100 ha for each mange zone with available data. All populations of each mange zone were monitored in summer (July) by a single census day every other year.

To account for years in which census was not carried out, the acquired dataset was filled in by calculating the arithmetic mean of the two adjacent data. Thematic maps were produced from population data for the 2006–2020 period using Quantum GIS software (Qgis-Tethys) ver. 3.16.9 [18].

### 2.3. Ordinary Surveillance Data

Ordinary surveillance was carried out by Trento Forestry Department, gamekeepers, PPSM park wardens and hunters from 2006 to 2020 during their daily activity of territorial supervision. The involved personnel have been trained on case detection and carcass handling; however, time and area dedicated to this activity were selected on the basis of a general organization and not specifically aimed at mange monitoring only.

For alive chamois, each individual showing both pruritic behaviour (scratching and gnawing) and alopecia was reported as a mange case.

For each dead chamois found in the field, the following data were recorded: date, sex, age, cause of death (natural or euthanasic/sanitary culling by wardens or trained hunters), locality name or geographic coordinates, management unit, mange zone and, when available, the results of parasitological diagnosis with the detection of *S. scabiei* mites in skin scrapings or in skin digested with potassium hydroxide (KOH) performed at the Istituto Zooprofilattico Sperimentale delle Venezie (IZSVe) Trento laboratory. Dead chamois were classified by tooth eruption and horn rings number [19] into three age classes: kids, yearlings and adults. If age estimation was not reliable, an “undetermined” age class was assigned.

A mange index case was defined as the first reported and laboratory confirmed scabietic carcass found in each mange zone.

To estimate the effective impact of mange versus other mortality causes, dead chamois were categorized into three mortality classes:Mangy: typical skin lesions confirmed by laboratory diagnosis of *S. scabiei*;Not mangy: no skin lesions, or suspect lesion not confirmed at the laboratory as *S. scabiei*;Undetermined: poor conservation status—unreliable diagnosis.

All data were geo-referenced by means of QGIS software for spatial analysis. Findings from 2006 to 2020 were represented in thematic maps according to the three mortality classes. Additionally, snow cover data registered by Passo Rolle meteorological station (46°17′49.88″ N, 11°47′15.32″ E) were recovered from www.meteotrentino.it (date of access: 11 January 2002).

### 2.4. Intensive Census and Enhanced Surveillance in CA

From 2008 to 2013, CA chamois population was investigated using an intensive census routine based on block count. Censuses were repeated every year, for five consecutive days, and were carried out by PPSN personnel, Trento Forestry Department and gamekeepers along six transects (length 1986 ± 1760 m), encompassing the entire area to minimize the probability of underestimation [20]. Moreover, for mange zones, chamois were assigned to six classes and the highest count, thus, obtained was assumed to be the minimum certain number of chamois in that particular year in order to calculate population density (n/100 ha).

From February 2008 to June 2010, an enhanced passive surveillance protocol was implemented in CA using the active involvement of gamekeepers, park wardens and hunters in a specifically designed monitoring activity aimed at the observer-initiated provision of relevant animal-health data [21]. According to this protocol, four transects were inspected every 10–15 days and traced in order to obtain maximum visibility over the entire area to identify mange cases in both alive and dead animals. Three were traced in the southern part of the area and monitored all year round while the northern one, not encompassing chamois wintering zones, was monitored only in the summer season.

All carcasses collected in CA using the intensive monitor protocol were delivered to the IZSVe Trento laboratory for diagnosis confirmation.

### 2.5. Data Analysis

In order to evaluate the impact of the mange epidemic wave on population dynamics, the demographic decline rate (Δ) was calculated for each mange zone by comparing the post-epidemic stock with the pre-epidemic one. The end of the epidemic wave was defined as the year with the lowest count during the period following the index case and characterized by a continuous detection of mange cases (no more than one year without cases). The Δ was estimated as the demographic decrement between the pre-epidemic (last count available before the index case) and the post-epidemic stock (lowest count at the end of the epidemic wave).
Δ = [(pre-epidemic stock − post-epidemic stock)/pre-epidemic stock] × 100.(1)

The same calculation was applied to CA.

In order to evaluate the performances of the enhanced surveillance protocol, the capacity to achieve a reliable mange status was estimated as the percentage of carcasses classified as “mangy” or “not mangy” out of the total carcasses retrieved and compared with the results of the ordinary surveillance by a 2-sample z-test, considering the period 2008–2010 for both systems. In addition, the capacity of the two surveillance systems in detecting as many dead animals as possible (sensitivity) was evaluated by comparing the respective number of carcasses not detected by surveillance. This was estimated by deducting the number of retrieved carcasses from the estimated number of dead animals, considering the period from the index case to the post-epidemic, for every mange zone and for the CA. Moreover, results were compared by the 2-sample z-test. Statistical tests were performed using Epitools (https://epitools.ausvet.com.au/ (date of access: 30 March 2022), keeping a *p*-value < 0.05 as significant.

## 3. Results

### 3.1. Description of the Mange Epidemic Using Ordinary Surveillance Data

The densities from 2006 to 2020 are shown in Figure 2. The decline rate of each mange zone is reported in Table 1 and Appendix A.

The first mange record inside the study area occurred in 2005 (PAN), while the index case inside the PPSM was registered in October 2007 (SCA). The epidemic wave moved along the different mange zones of the study area, showing a variable impact on the population. For each mange area, a negative trend in population was observed after the index case, although the time to reach the lowest value was very variable (from two to ten years). Moreover, the impact in terms of demographic decline ranged extensively by zone, from 19% (LSO) to 84% (VFS).

Due to the impact of mange in each mange zones and based on the mange management strategy, hunting was banned (strategy Type 5) for a minimum of 2 (PAN) to 13 years (ROL). In two mange zones (PAN and CAU), hunting was reintroduced (Type 7) by considering demographic recoveries following the post-epidemic situation. The summary of mange strategy applied in the different years in each mange zone is reported in Appendix A. 

According to ordinary surveillance data, 609 chamois were found dead or euthanized from 2006 to 2020 in the study area (Figure 3, Table 2 and Appendix A). Of these, 263 were males (40%), 218 were females (35%) and 128 were undetermined (25%). Concerning age, 57 were kids, 50 were yearlings, 436 were adults and 66 were undetermined. The mange’s status was determined in 406 (67%) animals, whereas in 203 (33%), it remained undetermined. Among the former, 330 (81%) were classified as “mangy” and 76 (19%) were classified as “not mangy”. Of these, 172 were males (52%), 132 were females (40%) and 26 were undetermined (7%). 

As is clear from Figure 4, the proportion of undetermined cadavers was higher in the early stages of the study and particularly during the 2009 peak when abundant winter precipitations hindered surveillance activities and delayed the finding of sufficiently well-preserved carcasses.

The epidemic wave moved from the northern and central zones (PAN, PAL, SCA and CAU) towards both the west (LIT, STE and LSO) and south (TOT, VFS and VFM). Interestingly, the epidemic wave took 4–5 years to reach southern zones from the initially affected central ones, which was probably due to the deep depression separating Vette Feltrine from the rest of area acting as a geographical barrier, which hampered migration between populations. The apparent bimodal shape of the number of chamois found dead (Figure 4), with a first mortality peak in 2009 (involving PAN, PAL, SCA and CAU) and a second in 2017 involving STE, VFS and VFM (Table 2, Figure 3 and Appendix A), is difficult to explain. The two peaks are clearly due to the spread of the mange from northern and central zones towards west and south, although an exceptionally high snowfall most probably contributed substantially to the first one (Figure 4). However, it is unclear why, in the years between the two peaks (particularly 2012–2015), very few mangy carcasses were found in some zones affected by the epidemic wave (i.e., LIT, LSO and TOT), suggesting a reduced sensitivity in the case detection of the ordinary passive surveillance in these areas. A comeback of a few mange cases several years after the first wave was detected in PAN (2018) and in LIT (2019) and in single sporadic cases in other two zones (ROL and TOT). However, we did not detect the evidence of a second wave at present, and in some zones, chamois populations recovered to pre-epidemic levels (Figure 2).

The results of the time series analysis of adult males and females are shown in Figure 5. For both, a seasonal pattern was observed with the highest case frequency in spring months (March–May), and the lowest one in late autumn–winter months (September–December). 

### 3.2. Description of Mange Epidemic in the CA Using Enhanced Surveillance Data

Results of intensive census for CA are reported in Table 3. The index case occurred in March 2008. As a census was not conducted with an intensive approach during 2007, it was not possible to use original data: We, therefore, assumed the number of animals counted in 2008 as “pre-epidemic stock”, added up with the number of chamois (*n* = 30) found dead in the period from the index case (March 2008) to the beginning of intensive census dates (July 2008), assuming these deaths as extraordinary mortality that is mainly mange-induced. The year 2011 was kept as post-epidemic stock, when the lowest number of chamois was counted. The estimated decline rate between 2008 and 2011 was 77.3%. In 2012, a new demographic increment was recorded.

The enhanced surveillance conducted from 2008 up to 2010 in CA allowed the detection of 73 chamois carcasses. The majority of animals included adults (22 females, 12 males and 6 undetermined), and only two were yearlings and one was a kid. For the remaining 29 carcasses, it was not possible to assign age, class nor sex. Overall, for 54 carcasses (74%), it was possible to determine mange statuses, showing mange in 34 (63%) cases.

The seasonality of the mange cases coincided with that of the general study area (Figure 6), with a clearer trend for “not mangy” dead animals found mostly during late spring (March–May).

### 3.3. Comparison between Ordinary and Enhanced Surveillance

Considering the period 2008–2010, the enhanced surveillance was able to determine the mange status in 74.0% (54 determined; 19 undetermined) of the cases, performing significantly better than 57.7% (128 determined; 94 undetermined) of ordinary surveillance (*z*-value = 2.5; *p* = 0.013).

The estimated pre- and post-epidemic stocks of chamois in each mange zone and in CA are reported in Table 4, together with the number of carcasses found during the epidemic wave (from the index case to the year with the lowest count) and the resulting percentage of carcasses not detected by passive surveillance. The latter varied considerably among mange zones, and the same proportion is significantly different between the entire study area managed by ordinary surveillance (77.9%) and the enhanced surveillance implemented in CA, where the estimated proportion (37.8%) is remarkably lower (*z*-value = 9.9; *p* < 0.001).

## 4. Discussion

Flanking the classical description of an epidemic wave of sarcoptic mange with the results of an enhanced passive surveillance system, although applied in a limited area (CA), allowed an improved framing of the relevance of this disease for the chamois population and an improved understanding with respect to the limits of passive surveillance.

The setup of a passive surveillance system before the arrival of mange in the study area paved the way for the early detection of the index case and the consequential monitoring of the evolution of mange outbreaks in a naïve chamois population. The annual densities (Table 1, Figure 2, Appendix A) showed a general demographic decrease, from values typical of a balanced population (6–10 chamois/100 ha) [19,22] to far lower values, particularly in 2013–2015. This trend was highlighted in almost all mange zones.

The monthly distribution of the overall mortality is comparable to previous results [13,23], identifying the seasonal case peak in spring (March–May, Figure 6) following the re-emergence of carcasses after snow melting, and the improving conditions for field work. This distribution is the same in all mange zones that present a number of significant findings, as well as in CA, monitored using the enhanced passive protocol.

Similar percentages of males and females were affected by mange in our study, confirming previous observations [8,13,23]. Regarding the age distribution of mange cases, the sample was not considered representative, as the carcasses of young animals were often almost completely consumed by necrophagic animals and are, therefore, unavailable for sampling and subsequent diagnostic analyses.

Overall, our results confirm the role of mange as a determining factor for chamois mortality and population dynamics, as already described in the literature [8,13], considering that more than half of the deaths (54.1%, Table 2) were due to mange, in the entire period of 2006–2020. At the same time, the data collected using an intensive census and enhanced surveillance in CA stressed the importance of other factors affecting population dynamics, as evidenced by the strong decline in the population observed in 2009, which was probably linked not only to sarcoptic mange but also to the exceptional snowfalls recorded in the winter between 2008 and 2009. This finding is supported also by the high number of dead carcasses not attributable to mange (*n* = 13) out of the 34 carcasses found overall in 2009 in CA (Table 3). This finding allowed an improved interpretation of concurrent peak of mortality recorded for the entire study area in 2009 and only partially attributable to mange (Figure 4), indicating that the “winter factor” played a significant role in increasing disease mortality. The first snowfalls usually take place in the late autumn–early winter, when young chamois are approximately five months old and are starting to wean. Therefore, exceptional events create hostile conditions in terms of food availability, cold-induced stress, higher energy consumption and increased susceptibility to parasitic infection, finally leading to huge mortality among chamois and particularly among kids. In CA, only 5 juveniles were counted in 2009, out of 37 kids counted in the previous year, supporting the existence of a correlation between age and mortality. This observation is in line with other studies [24,25,26,27,28] that identify the winter harshness as the most important variable for survival of wild ungulates. However, the relative role of factors influencing animal abundance along the time, such as pathogens [24,25] and extreme climatic events [26,27], is hard to disentangle. To improve our interpretation of this complex realities, the modelling and simulation approach can provide support in forecasting and comparing the impact on wild ungulate populations on different factors, including pathogens [25].

In naïve populations of alpine Caprinae, fine-tuned host–parasite regulation has yet to be established for *S. scabiei*; in such a situation, sarcoptic mange, at least in the early years, may act more likely as an unpredictable event rather than as a regulating phenomenon. Such an unpredictable factor is also reflected in high variabilities among mange zones reported in our study for the epidemic wave duration (1–9 years), for the decline rate (19–84%), for the proportion of carcasses not detected by surveillance (26–98%) and also for the time span from the index case to the mortality peak. At the same time, such a variability is probably also related to the limits of the ordinary passive surveillance, which mainly relies on the resources available to local stakeholders (e.g., number of game wardens and local hunters) as well as the accessibility of the area. The very limited number of carcasses found in some mange zones (i.e., ROL, LIT, LSO and TOT), associated with a clear decline in the population stocks, resulted in very high percentages of dead animals that were not detected by surveillance (Table 4: 89.5–98.4%), indicating that the sensitivity in cases of the detection of ordinary surveillance in these areas was low. Consequently, the numbers of dead animals are probably highly underestimated and suffer of a wide range of errors, suggesting precaution in their interpretation.

The use of an enhanced passive surveillance methodology may partly overcome the above-mentioned limitations. Our results revealed that the performances of the intensive approach were significantly higher both in terms of the identification of the cause of death (74.0% vs. 57.7%) and the reduction in the number of carcasses not detected by surveillance (37.8% vs. 77.9%). The more intensive field work applied in the CA was actually more effective in detecting dead animals in the field, allowing the discovery of well-conserved carcasses; the improved preservation of the skin led to significantly higher probabilities of reaching an etiological diagnosis. An enhanced surveillance protocol is able, therefore, to provide more reliable results compared to the ordinary protocol, although its application in a wider area requires a higher level of resources. Enhanced surveillance methods require the use of two dedicated personnel for a half-day to be multiplied for four transects for 2- or 3-times a month and for a total of 8–12 working days. As a matter of comparison, the ordinary surveillance implies one or two transects each month for a total of 1–2 working days. On average, the enhanced surveillance method requires about a tenfold amount of time dedicated by professional personnel, which obviously results in the need for the allocation of more resources. 

Another limitation present in our study and more generally in chamois population census methodology is linked with density estimation, which is often approached by considering the surface of the overall study area instead of focusing on the effective distributional range of the animals inside it. As a consequence, it is difficult to compare areas in which chamois populations are differently distributed. Consequently, implementing more efforts in the identification of chamois’ distributional range is essential, particularly in reference to the census period, since this aspect is likely to become increasingly important in the near future in light of the extensive changes in chamois’ ecology; in particular, factors such as climate change [29] and interspecific competition with red deer [30] should be considered.

Based on the above considerations, it appears evident that both census and dead animal data should be applied with caution for the decision on the action to be taken (e.g., hunting ban). As an example, in some mange zones of our study, the population declined by far under the threshold for mange transmission (1.8 chamois/100 ha), as indicated by a previous study [13]. Specifically, a minimum value of 1.76 chamois/100 ha was reached in ROL (2013), 1.21 chamois/100 ha in SCA (2015), 1.25 chamois/100 ha in PAL (2013), 1.34 chamois/100 ha in VFM (2018) and 0.44 chamois/100 ha in VFS (2019), suggesting that the uncertainty around these determinations may be high.

Because of the limitations of the passive surveillance, in both ordinary and enhanced applications, other monitoring strategies should be considered in the future as important assets to study the eco-epidemiology of this disease in wild Caprinae. In particular, besides deep investigations on the traits determining the immunopathology of the disease and possible genetic resistance/resilience [31], a reliable immunodiagnosis could allow active surveillance programs on hunted chamois aimed at estimating the introduction and spreading patterns of *S. scabiei*, its true prevalence and the proportion of clinical cases compared to the prevalence. For this purpose, activities are in progress in the area to apply to the Alpine chamois an ELISA tests validated in Iberian ibex showing a sensitivity of 100% and a specificity of 98.6% [32].

## 5. Conclusions

Our study summarises data collected during a 15-years activity aimed at an optimal management of a mange outbreak in the study area, providing interesting outputs in terms of the practical implementation of a passive surveillance approach and its consequential effects on hunting management.

Ordinary passive surveillance was demonstrated to be a useful tool for monitoring long-term demographic trends in an ample area, but it was also demonstrated to have a high variability among mange zones and generally a low sensitivity in case detection. 

The enhanced surveillance protocol implemented for few years in a small area (CA) demonstrated to be able to detect a higher number of dead animals and to improve the proportion of identification of the cause of death. The findings of the enhanced surveillance confirmed the preeminent role played by sarcoptic mange in chamois populations and the sharp decrease in CA, such as in many other mange zones. At the same time, it highlighted the important contribution of the exceptional snowfalls in the winter of 2008–2009. 

The early detection of an epidemic outbreak and the appropriate description of mortality and demographic trends in smaller areas (such as our mange zones) would benefit from the more intensive approach, which showed to be effective in CA, but its application in a wide area (such as our study area) may be too costly if applied for longer periods when considering that it requires a tenfold-higher amount of time dedicated by professional personnel.

## Figures and Tables

**Figure 1 animals-12-02077-f001:**
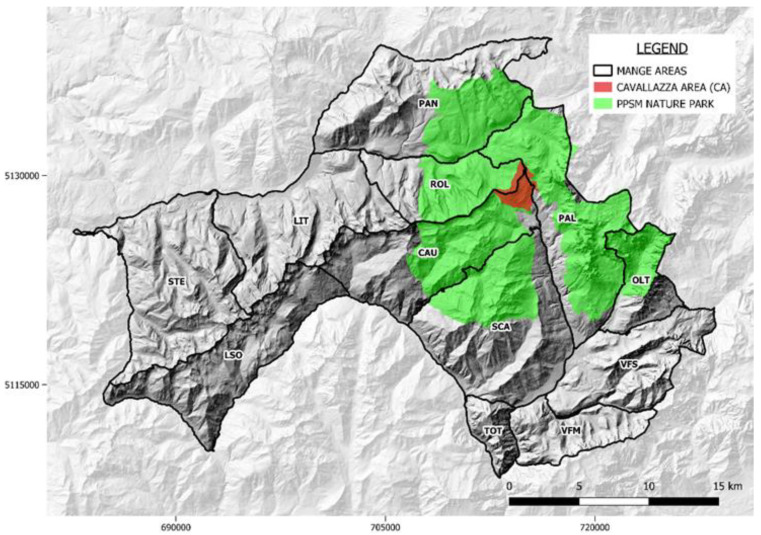
The study area divided by mange zones. CAU = Cauriol-Val Cigolera; LIT = Val Maggiore-Litegosa; LSO = Lagorai Sud Occidentale; OLT = Oltro; PAL = Pale; PAN = Paneveggio; ROL = Val Maggiore-Rolle; SCA = Scanaiol-Boalon; STE = Val Moena-Lagorai-Stelune; TOT = Totoga; VFM = Vette Feltrine Meridionali; VFS = Vette Feltrine Settentrionali. Cavallazza Area (CA) is represented with red color; the territory of the PPSM Nature Park is represented with green color.

**Figure 2 animals-12-02077-f002:**
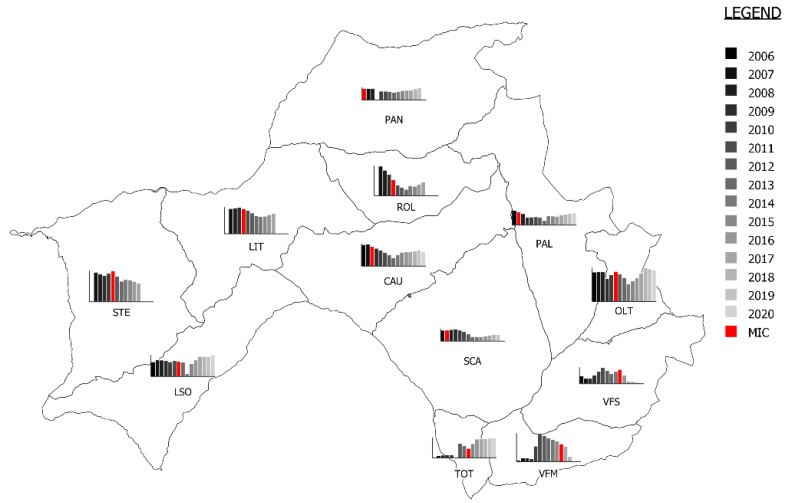
Trend in population density (grayscale). The year of the mange index case (MIC) is represented in red.

**Figure 3 animals-12-02077-f003:**
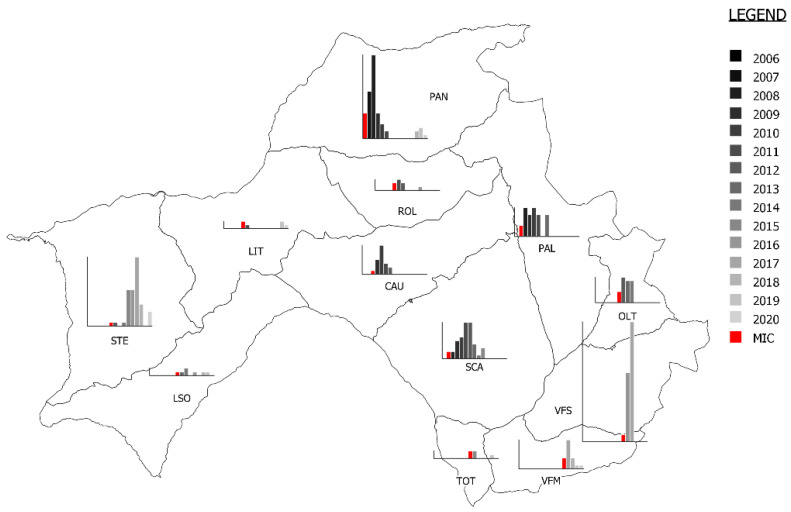
Trend in the number of mange cases (grayscale) detected through ordinary surveillance. The year of the mange index case (MIC) is represented in red.

**Figure 4 animals-12-02077-f004:**
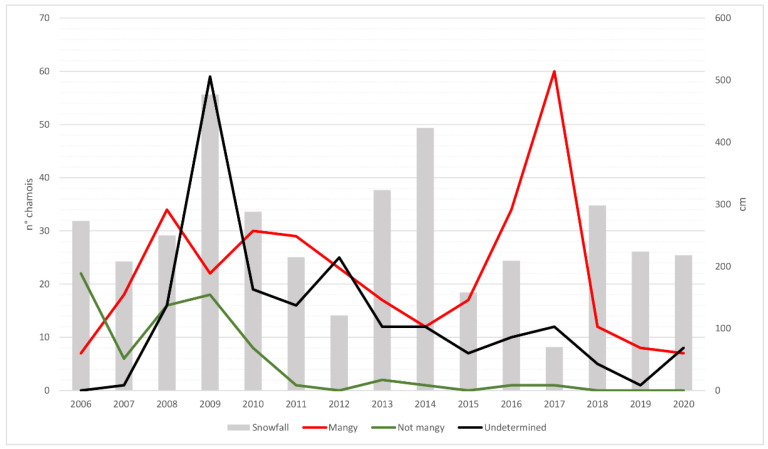
Number of dead chamois collected from 2006 to 2020, classified by the cause of death. Bars represent annual total snowfall in centimeters.

**Figure 5 animals-12-02077-f005:**
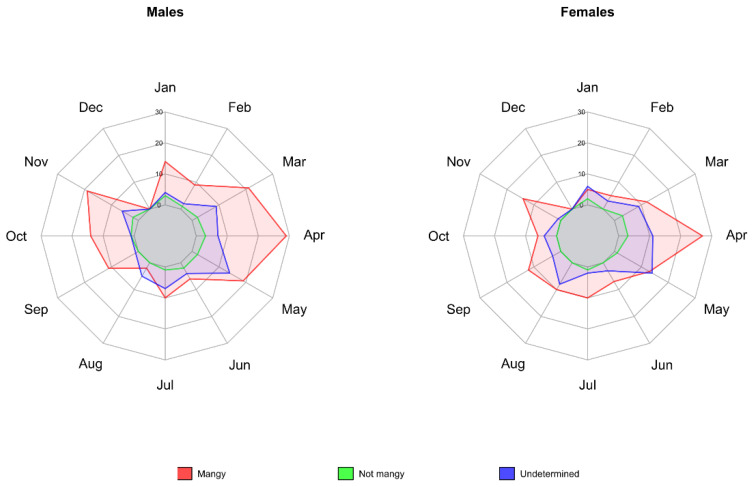
Monthly distribution of male and female chamois carcasses (2006–2020) found in the entire study area using ordinary passive surveillance, classified by mange status. Animals for which sex could not be assigned are not represented.

**Figure 6 animals-12-02077-f006:**
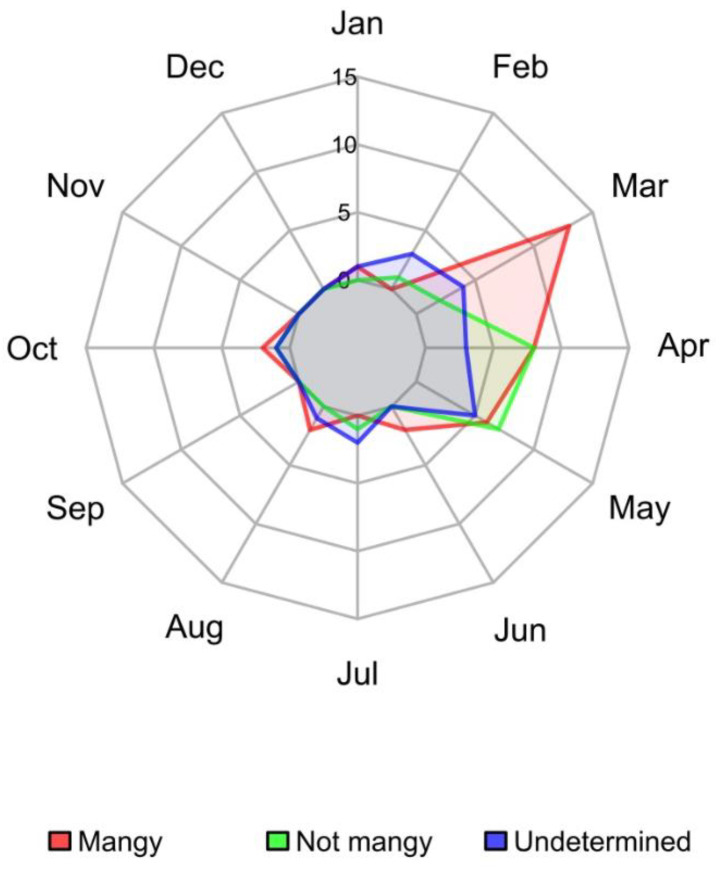
Monthly distribution of chamois carcasses (2006–2020) found in CA during enhanced surveillance, classified by mange status.

**Table 1 animals-12-02077-t001:** Index case, duration of the epidemic wave and decline rate calculated for each mange zone.

Mange Zone	Surface (ha)	Index Case	Pre-Epidemic Stock ^1^		Post-Epidemic Stock ^2^ (Census Year)		Epidemic Wave Duration (Years ^3^)	Decline Rate
			N Heads	Density	N Heads	Density		
PAN	9504.737	2005	342	3.60	231 (2011)	2.43	7	32%
PAL	7579.657	2007	313	4.13	95 (2013)	1.25	7	69%
SCA	9285.647	2007	277	2.98	112 (2015)	1.21	9	60%
CAU	7154.198	2008	434	6.07	165 (2013)	2.31	6	62%
ROL	3909.907	2010	232	5.93	69 (2013)	1.76	4	70%
LIT	7434.002	2010	543	7.30	356 (2014)	4.79	5	34%
STE	8852.382	2011	639	7.22	397 (2020)	4.48	10	38%
OLT	2679.183	2011	172	6.42	133 (2014)	4.96	4	23%
LSO	6595.381	2012	288	4.37	232 (2015)	3.52	4	19%
TOT	1321.142	2014	54	4.09	35 (2014)	2.65	1	35%
VFS ^4^	5282.734	2015	148	2.80	23 (2019)	0.44	5	84%
VFM ^4^	3057.262	2016	173	5.66	41 (2018)	1.34	3	76%

^1^ The value corresponds to the last census available before the year of the index case. ^2^ The value corresponds to the year with the lowest count during the period following the index case and characterized by a continuous detection of mange cases (no more than one year without cases). ^3^ Number of years from the index case to the year with the lowest count. In 2020, few mange cases were still reported in PAN, LIT, STE and VFM, but these cases represent a potential comeback or the last cases of the epidemic wave, and related chamois populations already started their stock recovery. ^4^ The Vette Feltrine zone was split into VFS and VMS in 2011.

**Table 2 animals-12-02077-t002:** Dead chamois found during ordinary surveillance by mange zone and year (number of mangy cases into brackets) and % of mangy cases out of the total dead chamois.

Mange Zone	2006	2007	2008	2009	2010	2011	2012	2013	2014	2015	2016	2017	2018	2019	2020	TOTAL	Percentage of Mange Cases
PAN	8 (7)	13 (13)	26 (23)	26 (7)	5 (4)	4 (2)	0	1	0	0	0	0	3 (2)	3 (3)	2 (1)	91 (62)	68.1%
PAL	1	4 (3)	11 (8)	25 (6)	12 (8)	10 (6)	1	8 (6)	1	0	1	0	2	0	0	76 (37)	48.6%
SCA	10	4 (2)	8 (2)	17 (5)	15 (6)	16 (10)	23 (10)	7 (4)	8 (1)	3 (3)	1	3	0	0	1	116 (43)	37.1%
CAU	6	1	9 (1)	19 (4)	18 (8)	6 (3)	10 (2)	1	1	1	0	1	0	0	1	74 (18)	24.3%
ROL	0	0	6	7	2 (2)	3 (3)	2 (2)	2	0	0	1 (1)	0	1	1	0	25 (8)	32.0%
LIT	0	0	0	0	2 (2)	1 (1)	0	0	0	0	0	0	0	2 (2)	1 (1)	6 (6)	100%
STE	0	0	0	0	0	1 (1)	1 (1)	0	1 (1)	10 (10)	10 (10)	19 (19)	6 (6)	0	4 (4)	52 (52)	100%
OLT	1	1	2	1	1	3 (3)	8 (7)	8 (6)	8 (6)	1	0	1	0	0	0	35 (22)	62.8%
LSO	2	0	0	0	0	0	1 (1)	1 (1)	2 (2)	0	1 (1)	1	1 (1)	1 (1)	2	12 (7)	58.3%
TOT	0	0	0	0	1	0	0	2	2 (2)	3 (2)	0	2	0	1 (1)	1	12 (5)	41.6%
VFS	0	2	2	1	0	1	2	0	2	5 (2)	27 (19)	38 (33)	1	0	0	81 (54)	66.6%
VFM	1	0	2	3	1	1	0	1	0	1	4 (3)	8 (8)	3 (3)	1 (1)	3 (1)	29 (16)	55.2%
TOTAL	29 (7)	25 (18)	66 (34)	99 (22)	57 (30)	46 (29)	48 (23)	31 (17)	25 (12)	24 (17)	45 (34)	73 (60)	17 (12)	9 (8)	15 (7)	609 (330)	54.1%

**Table 3 animals-12-02077-t003:** Results of the census of CA with classification by age and gender and the number of found carcasses in the last column. The data displayed for the year 2008 are corrected by adding up the number of chamois counted at the intensive census in July with the number of chamois found dead (brackets).

Year	Kids	Juveniles	Female Adults	Male Adults	Undetermined	Tot	Found Carcasses (Mangy- Not Mangy-Undetermined)
2008	37 (1)	21	60 (10)	21 (4)	15 (15)	154 (30)	34 (20-5-9)
2009	24	5	34	15	0	78	34 (11-13-10)
2010	10	10	15	9	0	44	5 (3-2-0)
2011	8	4	11	12	0	35	-
2012	22	10	24	8	0	64	-
2013	13	10	18	15	0	56	-

**Table 4 animals-12-02077-t004:** Number of found carcasses, estimated number of dead animals and percentage of animals not detected by passive surveillance in the mange zones and in the CA.

Mange Zone	Pre-Epidemic Stock	Post-Epidemic Stock	Epidemic Wave Duration	Number of Carcasses Found	Estimated Minimum Number of Dead Animals	Estimated Number of Animals Not Detected by Surveillance	Percentage of Dead Animals Not Detected by Surveillance
	N Heads	N Heads	Years				
PAN	342	231	7	82	111	29	26.1%
PAL	313	95	3	40	149	109	73.2%
SCA	277	112	9	101	165	64	38.8%
CAU	434	165	6	63	269	206	76.6%
ROL	232	69	4	9	163	154	94.5%
LIT	543	356	5	3	187	184	98.4%
STE	639	397	7	42	177	135	76.3%
OLT	172	133	4	27	39	12	30.8%
LSO	288	232	4	4	56	52	92.9%
TOT	54	35	1	2	19	17	89.5%
VFS	148	23	5	71	125	54	43.2%
VFM	173	41	3	15	132	117	88.6%
Ordinary surveillance (All mange zones)	3964	1889		459	2075	1616	77.9%
Enhanced surveillance (CA)	154	35	4	74	119	45	37.8%

## Data Availability

All relevant data are within the manuscript and its Appendix A files.

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
