# Peer review of "Description of a Sarcoptic Mange Outbreak in Alpine Chamois Using an Enhanced Surveillance Approach"

_animals, 2022, doi:10.3390/ani12162077_

Round 1
Reviewer 1 Report
The manuscript with ID: animals-1805990 aims to describe an outbreak of sarcoptic mange in Alpine chamois, as well as to compare two types of surveillance applied during the study period. The authors compile surveillance data over a 15-year period, from multiple subpopulations in the Dolomites, to provide descriptive data on population trends, trends on chamois found dead and chamois carcasses diagnosed with mange. In my opinion, the article does not add significant knowledge to the fields of mange nor surveillance, and could significantly improve with further work and by analyzing their data using other methods. For example, instead of presenting descriptive data (not new knowledge), the authors could use statistical modeling to infer the importance of mange in population growth during different phases of the outbreak dynamics, accounting for other known and significant sources of mortality, like snowfall, extreme climatic events, or hunting. These other sources of mortality could also affect the length of the “epidemic” period defined by the authors, and therefore affect the results. The authors do not discuss what happened in chamois populations after the “epidemic” period, did mange fade out from all chamois subpopulations? Was it maintained in an endemic occurrence? This study also doesn’t include any information about other susceptible hosts in the area, such as the Alpine ibex, which could also be involved in the different mange dynamics described. Overall, in my opinion, the epidemiological description and assessment needs to be improved for scientific interest and soundness.
The study also does a poor job in comparing different passive surveillance approaches, and it is not clear what is the real benefit of an enhanced passive surveillance for managers. Did the enhance surveillance improved or provided new knowledge? The authors do not properly assess what was the contribution of mange in population declines, therefore it is not clear why the enhanced surveillance was used. The authors could also improve the significance of the study by comparing other aspects of the two surveillance methods. Although the labour costs of the different types of surveillance is discussed and included in the abstract and conclusion, the study does not compare economic costs or working hours. The authors could also disclose results about who reports mange cases, possible contributions of general public, how geography affected surveillance in different areas, etc to make more interesting the study, as the current version does not provide much novel knowledge.
Overall, the writing should be improved in clarity in different sections, and the authors should be consistent with terminology throughout the manuscript. For example, the authors used systematic, intensive, routinary or ordinary surveillance in different parts of the text. I strongly suggest the authors to use well established terms in english for different types of surveillance, as I believe enhanced passive surveillance is more appropriate than “intensive” (please read Hoinville et al., Prev Vet Med 2013, 112:1-12, Ryser-Degiorgis, BMC Veterinary Research 2013, 9:223 or Artois et al., 2009, 187-213, in R.J. Delahay et al. (eds.), Management of Disease in Wild Mammals). The introduction does not provide a proper background on what is known about the epidemiology or the impact of mange in chamois or other mountain ungulates, and lacks of a section discussing types of surveillance, purposes, and what have been previously used to follow outbreaks in chamois or other mountain ungulates. I suggest rewriting the introduction and review the whole manuscript for clarity and English grammar. In my opinion, this manuscript requires substantial work from the authors and further analyses to provide a significant contribution to its field.
Few specific comments:
Line 45: sarcoptic mange has also been commonly associated with population declines in other Caprinae species, for example, in the Iberian Ibex. I suggest editing the sentence.
Line 63-73: the measures described here are not clearly explained and it leads to more questions than answers. For example, what range of hunting strategies were implemented? Where the same in all zones? What is a systematic passive surveillance? What methods for block census were implemented? Because this is part of the study, I suggest explaining properly all these measures in the material and methods section and simplify this part of the introduction.
Line 73: please remove “actually”.
Line 74: It is not clear to me what population, area or species are the focus of the surveillance. The structure of this paragraph is weird to me; I suggest the authors to edit this section to include upfront this information before talking about the specific objectives.
Line 78-83: this information should be provided before the objectives, please edit this last paragraph.
Line 85-100: in this section it should be described how the study area is subdivided, instead this information is spread out in the introduction. Overall, the way management units, study area, PPSM nature park and Cavalazza area are presented is very confusing and it should be improved.
Line 87: “, and it is entirely” please correct.
Line 98: please remove “also”.
Line 103: In the figure it is not clear what are the mange zones. The main text refers to management units, and not “pre-epidemic” management units, it is confusing.
Line 126: Please define who reports mange cases, the rangers? General public? Any user? Please state clearly where the passive surveillance was performed.
Line 125: No examination for S. scabiei was performed in other species than Alpine chamois?
Line 138: please describe better the methods used to detect S. scabiei from dead chamois, or provide a reference.
Line 143: I suggest writing “laboratory” instead of “lab”.
Line 152: I suggest introducing the abbreviation “CA” before in the text, and if possible, include it in the figure.
Line 169-172: the definition of “epidemic” made by the authors could be deduced by the text, but it is not stated or properly justified in the text. For example, I understand the authors assume that an epidemic length of a mange outbreak could be defined by population trends, but this assumption should be justified with references. It should also be justified why the authors did not use the detection of mange cases or other measures of disease presence and abundance.
Figure 192: I suggest including the abbreviations of the mange zones in the figure as well, it is difficult to relate with the information provided in the text.
Line 203: because some of the data is provided for the overall study area, it may be good to add a row in Table 2 with total numbers.
Line 220: Can you indicate the Primiero Valley in the map? The map in figure 1 could be improved to show more clearly the geography, by instead of using an orthophoto using hillshade options or/and altitudinal gradient colours.
Figure 221: “exchange of animals” is not a precise term, please correct.
Line 222: I believe the bimodal shape that the authors refer is to the chamois found dead and diagnosed with mange, but not necessarily the incidence of the disease or parasite, or as the authors said, the “epidemic”. I suggest correcting this sentence or clarify if the authors have other type of data to support this statement.
Line 225-227: This sentence is not clear, what do you mean by spatial trend? Perhaps an increase of transmission over a larger area? Please rewrite this sentence.
Line 225-227: According to Table 3 there are other mange zones in which mange was diagnosed in dead chamois, after several years of no detection. Can you clarify this?
Line 228: The structure of the results is a bit confusing to me, I suggest writing the results of mange diagnostics together with the results of chamois found dead in the previous paragraph.
Line 230: Figure 4 does not show percentages, please correct the sentence.
Line 231: what is the first phase of the study? I believe there was no previous description of “phases”, and it is confusing. Please define “phases” in the text if necessary and be consistent with terminology throughout the manuscript.
Line 231-233: this sentence is not clear, please rewrite.
Line 237: I suggest including the percentage of chamois found dead affected by mange in the total number row. Although this information is somehow present in the table, percentages provide a more straightforward information about the importance of mange as a cause of death.
Line 240: have you found seasonal differences on mange detection between adult males and females? If any, it would be good to split figure 5 by sex in adult animals.
Line 242: please complete the caption of Figure 5, what study areas, where are located, etc.
Line 248: what does it mean “integrated” in this sentence? I guess you added these animals, but please use non confusing terms that could be understood without the need of reading the whole manuscript.
Line 250-251: please rewrite this sentence.
Line 251: The term “rate” is usually used when comparing different periods of time, therefore an overall decline rate is confusing to me. Anyway, please specify the period in which you calculated the decline, it is not clear if it was an overall population decline of 77.3% during the whole epidemic period (I guess between 2007 to 2011) or an average decline rate of 77.3% per year.
Line 253: I suggest adding the number of chamois diagnosed with mange in this table.
Line 253-255: The numbers in brackets are not clear. The title of the table says that numbers in brackets are chamois found dead, but there are two numbers in brackets. Please correct.
Line 258: Please rewrite this sentence.
Line 242: The figure caption is incomplete, from where? Ordinary monitoring or surveillance? Please be precise and consistent, figure captions should be understandable by its own, without the need of reading the main text.
Line 267: Please complete the figure caption, what population or study area?
Line 276-277: Please change “lost” by “not detected by passive surveillance”.
Line 283: The population estimates are counts of animals in transects, which is an estimate with a range of error that may actually vary per year depending on several different factors. These absolute numbers also don’t account for the productivity of the population, therefore it is wrong to directly estimate from these the dead animals not detected by surveillance. I suggest to at least refer to “minimum N dead animals” and change consequently the other names in the table headings. It is also not clear, by reading the manuscript, if hunting was banned in all zones once the first mange case was detected, and until when. The limitations of this approach should be thoroughly discussed in the discussion section
Line 283: Please define “N”.
Line 283: Please correct the table title by “animals not detected by passive surveillance”. The table also includes other type of data, in my opinion the title is not enough precise or informative.
Line 288: What is a routine surveillance?
Line 302: I believe the study does not show results of males and females affected by mange. Please review.
Line 307: the results of this work do not properly assess the effect of mange epidemics in chamois population dynamics, as no effort to disentangle the effects of other sources of mortality was done by the authors.
Line 307: the reference num. 13 is not about chamois, please correct.
Line 308: Please be consistent with the terminology, do you refer to higher decrease rates? I believe these results by years are not presented properly in the manuscript.
Line 314-319: Snowfall and harsh winters also affects adult survival, please edit this section of the discussion accordingly, and add supporting references to each of your statements, rather than adding them all together at the end.
Line 320: I believe the term vitality is not widely used, and sounds that it may involve fertility? Please use well established terms or describe them in the text to avoid confusion.
Line 321: statistical modelling could be used to assess the amount of variability of population growth is explained by mange or by snowfall. Have the authors considered the use of other type of approaches?
Line 343-351: as the authors mentioned in the previous paragraph, the calculation of chamois density by simply dividing the total number of animals by the surface area is incorrect, and may lead to significant errors. Areas with more abrupt or not suitable habitat for chamois may bias density calculations. However, I do not agree with the authors that errors in chamois density calculations are associated with passive surveillance methods. Furthermore, the objective of passive surveillance should not be obtaining detailed epidemiology data, there are other approaches like targeted studies that may provide this type of data.
Line 358-361: please rewrite, for clarity and typos.
Line 365-380: the conclusion section contains part of discussion and provides poor conclusions from the work, not always related to the results presented in the study.
Author Response
Reviewer 1: Rebuttal letter
Comments and Suggestions for Authors - Reviewer 1
The manuscript with ID: animals-1805990 aims to describe an outbreak of sarcoptic mange in Alpine chamois, as well as to compare two types of surveillance applied during the study period. The authors compile surveillance data over a 15-year period, from multiple subpopulations in the Dolomites, to provide descriptive data on population trends, trends on chamois found dead and chamois carcasses diagnosed with mange. In my opinion, the article does not add significant knowledge to the fields of mange nor surveillance, and could significantly improve with further work and by analyzing their data using other methods. For example, instead of presenting descriptive data (not new knowledge), the authors could use statistical modeling to infer the importance of mange in population growth during different phases of the outbreak dynamics, accounting for other known and significant sources of mortality, like snowfall, extreme climatic events, or hunting. These other sources of mortality could also affect the length of the “epidemic” period defined by the authors, and therefore affect the results . The authors do not discuss what happened in chamois populations after the “epidemic” period, did mange fade out from all chamois subpopulations? Was it maintained in an endemic occurrence ? This study also doesn’t include any information about other susceptible hosts in the area, such as the Alpine ibex , which could also be involved in the different mange dynamics described. Overall, in my opinion, the epidemiological description and assessment needs to be improved for scientific interest and soundness.
The study also does a poor job in comparing different passive surveillance approaches, and it is not clear what is the real benefit of an enhanced passive surveillance for managers . Did the enhance surveillance improved or provided new knowledge? The authors do not properly assess what was the contribution of mange in population declines, therefore it is not clear why the enhanced surveillance was used. The authors could also improve the significance of the study by comparing other aspects of the two surveillance methods. Although the labour costs of the different types of surveillance is discussed and included in the abstract and conclusion, the study does not compare economic costs or working hours . The authors could also disclose results about who reports mange cases, possible contributions of general public, how geography affected surveillance in different areas, etc to make more interesting the study, as the current version does not provide much novel knowledge.
Overall, the writing should be improved in clarity in different sections, and the authors should be consistent with terminology throughout the manuscript. For example, the authors used systematic, intensive, routinary or ordinary surveillance in different parts of the text. I strongly suggest the authors to use well established terms in english for different types of surveillance, as I believe enhanced passive surveillance is more appropriate than “intensive” (please read Hoinville et al., Prev Vet Med 2013, 112:1-12, Ryser-Degiorgis, BMC Veterinary Research 2013, 9:223 or Artois et al., 2009, 187-213, in R.J. Delahay et al. (eds.), Management of Disease in Wild Mammals). The introduction does not provide a proper background on what is known about the epidemiology or the impact of mange in chamois or other mountain ungulates , and lacks of a section discussing types of surveillance, purposes, and what have been previously used to follow outbreaks in chamois or other mountain ungulates . I suggest rewriting the introduction and review the whole manuscript for clarity and English grammar. In my opinion, this manuscript requires substantial work from the authors and further analyses to provide a significant contribution to its field.
We are grateful to the reviewer for the detailed analysis of our manuscript, and for the useful comments. Our study summaries data collected during a 15-years activity aimed at an optimal management of a mange outbreak in a sufficiently wide area, providing interesting outputs in terms of practical implementation of a passive surveillance approach, and its consequential effects on the hunting management (this aspect is now improved in the main text and thanks to the inclusion of two new supplementary materials, with the aim of explaining how data collected were used as a decision tool for the wildlife management). At the same time, we believe that the study highlighted the limits of a passive surveillance approach. In this regard, one of the main novelties of our study is the application of an enhanced passive surveillance protocol in a small area, to investigate its outcomes. In the revised manuscript, we attempted to present and discuss with more clarity advantages and disadvantages of the ordinary and enhanced protocol (e.g. showing how the sound data collected in CA helped to identify the winter factor as an important contributing factor to the mortality peak in 2009 – lines 391-410 in the Discussion section)
To our opinion, the types of data collected in our study can hardly be analyzed through a statistical model, due to the high uncertainty around the number of retrieved carcasses in some of the mange zones that will influence the overall output. We commented this point in the revised manuscript and focused our analysis on the comparison between the enhanced and the ordinary passive surveillance, as previously explained, also to meet the request from the reviewer comment.
Finally, we revised thoroughly the whole manuscript to improve the clarity of the text and the consistency of the terminology. More details are provided in the answers to the specific comments here below.
Few specific comments:
Line 45: sarcoptic mange has also been commonly associated with population declines in other Caprinae species, for example, in the Iberian Ibex. I suggest editing the sentence .
We have added some examples of mange outbreaks in Europe
Line 63-73: the measures described here are not clearly explained and it leads to more questions than answers. For example, what range of hunting strategies were implemented? Where the same in all zones? What is a systematic passive surveillance? What methods for block census were implemented? Because this is part of the study, I suggest explaining properly all these measures in the material and methods section and simplify this part of the introduction.
We moved in the material and methods the applied measures. We added as supplementary material the mange management strategy in chamois populations of Trento Province (Table S1) and the summary of the outcomes of the application of this strategy for each mange zones (Tabel S2). The “block count” method for census is now described in details in the manuscript (127-136).
Line 73: please remove “actually”.
Removed
Line 74: It is not clear to me what population, area or species are the focus of the surveillance. The structure of this paragraph is weird to me; I suggest the authors to edit this section to include upfront this information before talking about the specific objectives.
Done as suggested
Line 78-83: this information should be provided before the objectives, please edit this last paragraph .
This paragraph was moved in the study area description, in the M&M section
Line 85-100: in this section it should be described how the study area is subdivided, instead this information is spread out in the introduction. Overall, the way management units, study area, PPSM nature park and Cavalazza area are presented is very confusing and it should be improved.
We moved here the whole description of the subdivision of the study area. Any reference to management units is now removed, since probably confusing and not very relevant (they were not deployed during the outbreak). The geographic description of the study area is now improved also thanks the changes made for Figure 1, according to a following comment..
Line 87: “, and it is entirely” please correct.
We added “it”
Line 98: please remove “also”.
Removed
Line 103: In the figure it is not clear what are the mange zones. The main text refers to management units, and not “pre-epidemic” management units, it is confusing.
We have removed any reference to management units.
Line 126: Please define who reports mange cases, the rangers? General public? Any user? Please state clearly where the passive surveillance was performed.
We added the sentence “Passive surveillance was carried out by Trento Forestry Department, gamekeepers, PPSM park wardens and hunters”
Line 125: No examination for S. scabiei was performed in other species than Alpine chamois ?
Yes also in the Alpine ibex, protected species in Italy. In particular, in the study area, the ibex population had been reintroduced few years before the mange introduction (year 2000) and suffered a population collapse due to sarcoptic mange.
Line 138: please describe better the methods used to detect S. scabiei from dead chamois, or provide a reference .
Visual diagnosis with the detection of mites under microscopic in skin scrapings or in skin digested with potassium hydroxide (KOH)
Line 143: I suggest writing “laboratory” instead of “lab”.
Done
Line 152: I suggest introducing the abbreviation “CA” before in the text, and if possible, include it in the figure.
Done
Line 169-172: the definition of “epidemic” made by the authors could be deduced by the text, but it is not stated or properly justified in the text. For example, I understand the authors assume that an epidemic length of a mange outbreak could be defined by population trends, but this assumption should be justified with references . It should also be justified why the authors did not use the detection of mange cases or other measures of disease presence and abundance.
We agree with the reviewer that the point of the epidemic duration was not sufficiently well presented in the text, mostly not to overload the M&M Section. The calculation of the duration of the epidemic wave in each mange zone was particularly complex to handle, due to the paucity of surveillance data coming from carcass retrieval in some zones. As a consequence, we opted for using mainly the demographic trend, using the lowest count in the period following the index case, after some consecutive years with population decrement. However, in the revised manuscript we modified our definition, which is taking into account the detection of mange cases. This definition is now better explicated in M&M Section (195-200. In this new definition, the demographic trend is integrated by the disease presence approach: “period following the index case and characterized by a continuous detection of mange cases (no more than one year without cases)”. To our opinion, using only mange case detection will underestimate the epidemic wave duration in the many zones, due to the low sensitivity for case detection of the ordinary surveillance. The calculation made in Tab. 2 and Tab. 5 has been revised and slightly modified for two mange zones (STE and PAL) and for the total. Finally, in the framework of the overall revision of the terminology, we changed the term “epidemic” in “epidemic wave”, avoiding to use the term “outbreak” for each single mange zone, since this term is already used in the title of the manuscript for the entire area.
Figure 192: I suggest including the abbreviations of the mange zones in the figure as well, it is difficult to relate with the information provided in the text.
Done
Line 203: because some of the data is provided for the overall study area, it may be good to add a row in Table 2 with total numbers.
The numbers of chamois heads and the associated density values refer to different years for each zone (that means for each row), and it’s therefore not correct to sum up these number in a row with the total. The main aim of this table is to highlight the differences in decline rates and in epidemic wave duration among zones.
Line 220: Can you indicate the Primiero Valley in the map? The map in figure 1 could be improved to show more clearly the geography, by instead of using an orthophoto using hillshade options or/and altitudinal gradient colours.
We changed the figure 1 with hillshade options as requested. We changed Primiero Valley with “deep depression separating Vette Feltrine from the rest”
Figure 221: “exchange of animals” is not a precise term, please correct.
We replaced with “…which hampered migration between populations”
Line 222: I believe the bimodal shape that the authors refer is to the chamois found dead and diagnosed with mange, but not necessarily the incidence of the disease or parasite, or as the authors said, the “epidemic”. I suggest correcting this sentence or clarify if the authors have other type of data to support this statement.
We deeply modified the whole paragraph al lines 216-227 (now lines 264-282 to improve its clarity (e.g., reporting the abbreviation of the mange zones, instead of local names, which can result unclear). The aim of the paragraph is to describe the movement of the mange wave, from the northern and central zones towards western and southern area. Only the southern ones were reached with more delay, and this is now corrected in the revised manuscript. In order to show better this spatial movements, the Table 2 has been modified reporting the mange zone in the rows according to the chronological appearance of the index case. For a matter of internal coherence also Table 1 and Table 4 have been modified accordingly. A new hypothesis on the bimodal peaks has been introduced here (lines 272-278, since the two peaks are clearly due to the spread of the mange from northern and central zones towards west and south, but it’s unclear why during this spread in some zones very few mangy carcasses were found. This point is also discussed in the Discussion section.
Line 225-227: This sentence is not clear, what do you mean by spatial trend? Perhaps an increase of transmission over a larger area? Please rewrite this sentence.
The whole paragraph has been rewritten, and the specific sentence deleted.
Line 225-227: According to Table 3 there are other mange zones in which mange was diagnosed in dead chamois, after several years of no detection. Can you clarify this?
The sentence has been revised (now lines 274-282)
Line 228: The structure of the results is a bit confusing to me, I suggest writing the results of mange diagnostics together with the results of chamois found dead in the previous paragraph.
We agree and moved this paragraph, after the sentence “Concerning age, 57 were kids, 50 yearlings, 436 adults and 66 undetermined.” This change is not highlighted in the revised manuscript.
Line 230: Figure 4 does not show percentages, please correct the sentence.
We replaced with “proportion”
Line 231: what is the first phase of the study? I believe there was no previous description of “phases”, and it is confusing. Please define “phases” in the text if necessary and be consistent with terminology throughout the manuscript.
We replaced with “the early stages of study”
Line 231-233: this sentence is not clear, please rewrite.
The sentence has been rewritten.
Line 237: I suggest including the percentage of chamois found dead affected by mange in the total number row. Although this information is somehow present in the table, percentages provide a more straightforward information about the importance of mange as a cause of death.
We agree with reviewer observation that percentages offer a more straightforward information, and we included a new column with the percentage of mange as cause of death for each zone. Again, we are more interested in the differences among zones, than along the years of the study.
Line 240: have you found seasonal differences on mange detection between adult males and females? If any, it would be good to split figure 5 by sex in adult animals.
We replaced with a new figure with adult males and females separated
Line 242: please complete the caption of Figure 5, what study areas, where are located, etc.
We specified in the caption of the new Fig 5 that graphs are referred to the whole study area (ordinary passive surveillance)
Line 248: what does it mean “integrated” in this sentence? I guess you added these animals, but please use non confusing terms that could be understood without the need of reading the whole manuscript.
We replaced with “added up” in the main text and “adding up” in the caption
Line 250-251: please rewrite this sentence.
The part into brackets has been reduced because redundant and a new sentence added at the end, to improve the clarity.
Line 251: The term “rate” is usually used when comparing different periods of time, therefore an overall decline rate is confusing to me. Anyway, please specify the period in which you calculated the decline, it is not clear if it was an overall population decline of 77.3% during the whole epidemic period (I guess between 2007 to 2011) or an average decline rate of 77.3% per year.
We are referring to the “demographic decline rate”, as defined at the beginning of the section “2.5. Data analysis”, therefore the decline for the whole period (2007-2011). We deleted the word “overall” and inserted the years for better clarity.
Line 253: I suggest adding the number of chamois diagnosed with mange in this table.
We included a new column with the number of found carcasses and, among these, the ones diagnosed as mange cases, as not mangy and as undetermined.
Line 253-255: The numbers in brackets are not clear. The title of the table says that numbers in brackets are chamois found dead, but there are two numbers in brackets. Please correct.
We corrected, reporting only the number of chamois found dead in brackets.
Line 258: Please rewrite this sentence.
The second part of the sentence has been deleted, since detailed data are now reported in the Table 3.
Line 242: The figure caption is incomplete, from where? Ordinary monitoring or surveillance? Please be precise and consistent, figure captions should be understandable by its own, without the need of reading the main text.
We specified in the caption that those are referred to the study area (ordinary passive surveillance)
Line 267: Please complete the figure caption, what population or study area?
We have specified in the caption that those are referred to CA (enhanced passive surveillance)
Line 276-277: Please change “lost” by “not detected by passive surveillance ”.
Done
Line 283: The population estimates are counts of animals in transects, which is an estimate with a range of error that may actually vary per year depending on several different factors. These absolute numbers also don’t account for the productivity of the population, therefore it is wrong to directly estimate from these the dead animals not detected by surveillance. I suggest to at least refer to “minimum N dead animals” and change consequently the other names in the table headings. It is also not clear, by reading the manuscript, if hunting was banned in all zones once the first mange case was detected, and until when. The limitations of this approach should be thoroughly discussed in the discussion section
We are aware of the limit of this calculation, and of the range of error surrounding both census and dead animal estimates. For this reason, we apply our calculation to the overall period only (and not to each single year), considering that the higher number can temper the uncertainty. We adapted the calculation from Rossi et al., 1995, from where we adopted the term “lost to surveillance”. However, we agree with reviewer terminological suggestions and correct the revised manuscript. As regard hunting, this point is now better explained throughout the whole manuscript and details can be found in Supplementary Material: the Mange management strategy (Table S1) and the summary of application of mange strategy for each mange zone from 2006 to 2020 (Table S2)
Line 283: Please define “N”.
N is Number, we changed to “Number “
Line 283: Please correct the table title by “animals not detected by passive surveillance”. The table also includes other type of data, in my opinion the title is not enough precise or informative.
We corrected the title according to reviewer suggestion.
Line 288: What is a routine surveillance?
We changed with ordinary surveillance, terminology introduced and explained in the M&M section.
Line 302: I believe the study does not show results of males and females affected by mange. Please review.
The information on number of males and females affected is now reported at line 257
Line 307: the results of this work do not properly assess the effect of mange epidemics in chamois population dynamics , as no effort to disentangle the effects of other sources of mortality was done by the authors.
We agree with the reviewer that our study did not demonstrate through statistical model the association between mange cases and demographic dynamics of chamois population (as previously explained, our data suffer of too much uncertainty to attempt a statistical model). However, more than half of the dead carcasses were attributed to mange, and we actually attempted to disentangle the effects of other causes, although the number of undetermined deaths was still high, mostly in the ordinary passive surveillance. A new sentence has been added in this point and the remaining text modified accordingly.
Line 307: the reference num. 13 is not about chamois, please correct.
We removed the quotation
Line 308: Please be consistent with the terminology, do you refer to higher decrease rates? I believe these results by years are not presented properly in the manuscript.
The sentence refers to date actually not presented clearly in the previous manuscript and it’s not much relevant to the discussion, therefore has been deleted.
Line 314-319: Snowfall and harsh winters also affects adult survival, please edit this section of the discussion accordingly, and add supporting references to each of your statements, rather than adding them all together at the end.
We modified the sentence, including adult, but maintaining a special reference to young ones, as demonstrated by our data from CA, which are now explicitely mentioned in this part of the Discussion. References were modified as requested
Line 320: I believe the term vitality is not widely used, and sounds that it may involve fertility? Please use well established terms or describe them in the text to avoid confusion.
We agree and substitute both terms with a more general “survival”.
Line 321: statistical modelling could be used to assess the amount of variability of population growth is explained by mange or by snowfall. Have the authors considered the use of other type of approaches?
As previously explained, the data from both census and passive surveillance can be considered precise and sound only in CA, whereas wide range of errors are associated with the ordinary surveillance, hampering their use in a statistical model.
Line 343-351: as the authors mentioned in the previous paragraph, the calculation of chamois density by simply dividing the total number of animals by the surface area is incorrect, and may lead to significant errors. Areas with more abrupt or not suitable habitat for chamois may bias density calculations. However, I do not agree with the authors that errors in chamois density calculations are associated with passive surveillance methods. Furthermore, the objective of passive surveillance should not be obtaining detailed epidemiology data, there are other approaches like targeted studies that may provide this type of data.
To our opinion, the density or consistency data of the population help in the interpretation and cross-checking of the passive surveillance data, as demonstrated by our study, and they are therefore somehow associated. We agree that a structural error in density calculation will not affect too much, since it will influence similarly all data (e.g., pre and post-epidemic stocks). However, we modified the first sentence and deleted the second one, considering that moving part of the Conclusion section immediately after this paragraph would make the second sentence redundant.
Line 358-361: please rewrite, for clarity and typos.
The whole paragraph has been moved before in the section, after the discussion on the limits of the ordinary surveillance (change not highlighted in the revised manuscript). The part mentioned by the reviewer has been deleted, because not much relevant to our results.
Line 365-380: the conclusion section contains part of discussion and provides poor conclusions from the work, not always related to the results presented in the study.
We agree with the reviewers and moved at the end of the Discussion the lines 270-380 of the original manuscript (change not highlighted in the revised manuscript). The rest of the Conclusion has been further implemented.

Reviewer 2 Report
Reviewer(s)’ General Comments to Authors:
This is an interesting paper and the topic is very useful. The manuscript is overall well written, informative and interesting. Hence, it can be considered for publication. However, there are some points which need to be improved or changed. I recommend this manuscript for publication in Animals, provided that the minor revision will have been made by the Authors in the re-edited and resubmitted version of the current paper according to the remarks of the Reviewer indicated below:
Line 39-84, The introduction needs to be shortened/more specific (it looks more like a discussion)
Line 61, …. As such…
Line 113, …zone was monitored…
Line 115, … Chamois was observed…
Line 135. … In case of age estimation…
Line 303, … was not considered representative…
Line 311, … after 2009, which was probably…
Line 315, … when young chamois is…
Line 326, … Such an unpredictability is…
Line 330, … such a variability is…
Line 338, … put more effort into the… rephrase
Line 341, … particulars…
Line 354, … approach was significantly higher…
Line 355, … reduction in the number…
Moreover, the conclusion could contain more comments to be more specific and clear for the reader of what your study showed eventually, with more detailed information. Some clarification and a greater focus on why the study was conducted with the impact of the findings will be very helpful in the short term at the conclusion.
Author Response
Reviewer 2 – Rebuttal letter
Reviewer(s)’ General Comments to Authors:
This is an interesting paper and the topic is very useful. The manuscript is overall well written, informative and interesting. Hence, it can be considered for publication. However, there are some points which need to be improved or changed. I recommend this manuscript for publication in Animals, provided that the minor revision will have been made by the Authors in the re-edited and resubmitted version of the current paper according to the remarks of the Reviewer indicated below:
Line 39-84, The introduction needs to be shortened/more specific (it looks more like a discussion)
Line 61, …. As such…
Done as suggested
Line 113, …zone was monitored…
Done as suggested
Line 115, … Chamois was observed…
We added “the” chamois were observed
Line 135. … In case of age estimation…
We changed with “if”
Line 303, … was not considered representative…
Done as suggested
Line 311, … after 2009, which was probably…
We changed
Line 315, … when young chamois is…
Done as suggested
Line 326, … Such an unpredictability is…
Done as suggested
Line 330, … such a variability is…
Done as suggested
Line 338, … put more effort into the… rephrase
We changed
Line 341, … particulars…
Done as suggested
Line 354, … approach was significantly higher…
We changed the sentence
Line 355, … reduction in the number…
We changed the sentence
Moreover, the conclusion could contain more comments to be more specific and clear for the reader of what your study showed eventually, with more detailed information. Some clarification and a greater focus on why the study was conducted with the impact of the findings will be very helpful in the short term at the conclusion.
Conclusion has been modified, as per suggestion also of reviewer 1.
Reviewer 3 Report
Thank you so much for your manuscript. I found it very interesting in your presentation; however, I suggest minor corrections to improve a little bit its understanding. You can see my comments below:
- Line 126: Could you provide more information about the diagnosis of mange cases for alive chamois? Do you think is enough for diagnosis the mange cases only based on clinical signs? The parasitological diagnosis should be used to confirm mange infestation.
- Line 132: What kind of parasitological diagnosis is used to confirm the infestation of sarcoptic mange?
- Line 137: Could you provide more information about microscopic evidence to confirm as mange cases?
- Line 143: What kind of lab diagnosis that used to confirm S. scabiei? Please clarify.
Author Response
Reviewer 3 – Rebuttal letter
Thank you so much for your manuscript. I found it very interesting in your presentation; however, I suggest minor corrections to improve a little bit its understanding. You can see my comments below:
Line 126: Could you provide more information about the diagnosis of mange cases for alive chamois? Do you think is enough for diagnosis the mange cases only based on clinical signs? The parasitological diagnosis should be used to confirm mange infestation.
No, it only provides an indication of animals with mangy-like lesions that must be confirmed with the finding of a carcass and confirmation of Sarcoptes scabiei through laboratory investigations
Line 132: What kind of parasitological diagnosis is used to confirm the infestation of sarcoptic mange?
See below
Line 137: Could you provide more information about microscopic evidence to confirm as mange cases?
See below
Line 143: What kind of lab diagnosis that used to confirm S. scabiei? Please clarify.
Visual diagnosis with the detection of mites under microscopic in skin scrapings or in skin digested with potassium hydroxide (KOH)
Round 2
Reviewer 1 Report
The reviewed version of animals-1805990 has improved substantially, and the authors have addressed or justified satisfactorily most of my previous comments. However, I have further comments that needs to be addressed before recommending this manuscript to be published. The authors should make some further efforts to organize better and clarify the information. For example, I do like the supplementary information they have provided, but the content in Table S1 is unclear. The authors did not disclose any information about the possibility of other ungulates being involved in the mange dynamics. Although I understand they want to focus on alpine chamois, in my opinion, they should state if other species were also detected with mange in the study area, and at least discuss if the Alpine ibex may have been involved in some of the dynamics described in chamois, for example, in those areas where sarcoptic mange was sporadically detected in chamois after few years of no detection.
The authors state in the conclusion that enhance passive surveillance is “too costly” to apply it in wide areas and for long periods, but the study does not provide any data about costs or labor hours of the two different types of surveillance, and it is not properly discussed in the text. In my opinion, providing some of this information, which should not be too difficult, would substantially improve the quality of the study as it is an important decision for managers. Nevertheless, the conclusions section is still poor, and I encourage the authors to rewrite it.
Specific comments:
Line 25: please correct “has been affected”
Line 34-35: it may not be appropriate to include in the abstract a statement like this when the study does not assess costs of the different surveillance approaches, and it has barely been discussed in the study. In addition, I don’t know what the authors mean by an “active approach” in passive surveillance. I suggest rewriting these sentences, this is to the discretion of the authors, but the abstract does not stretch enough the content of the study.
Line 41: Please review the manuscript for double or missing spaces, there is a missing space after the point.
Line 49: “for its conservation”
Line 50: please review sentence for English grammar.
Line 52: “conservation” seems out of place in this sentence and it has been discussed in previous sentenes, please correct.
Line 53: I suggest adding a reference, there are several examples in the literature already cited.
Line 54: Does “Sarcoptic” need to be capitalized? Similar for line 58 and elsewhere.
Line 55: “influenced by the interaction”. Please review carefully the whole manuscript for English grammar.
Line 56: is this statement supported by literature? It is a strong statement, the authors may want to phrase this sentence differently as mortality, in a longer temporal scale, also differ among naïve populations, as demonstrated by your own study.
Line 58: “naïve populations” or “unaffected populations” have more sense to me than “areas”.
Line 77: I believe “summaries” is incorrect.
Line 98: Does any of the reactive measures targeted the removal of diseased animals? For example, in the type 1 of hunting strategy? I think this is very important to clarify it as this has been a debate on how to manage mange outbreaks. Please state it clear.
Line 99: consider capitalizing each sentence.
Line 145: this sentence is unclear, please rewrite.
Line 153: for what I understood the rangers did not perform a mange control, but rather a mange case detection/monitoring, etc please correct or explain what the mange control was.
Line 154: please delete “association of”.
Line 168: “both study area and CA” Isn’t CA within the study area? This sentence is confusing, please correct.
Line 177: “Cavallazza Area (CA)” is not a clear subheading, and it is confusing because you have already described this area. Please correct it to describe what information you are including in this section. The authors may also consider moving the census of CA to the demographic data section, and dedicate two separate sections to describe the ordinary passive surveillance and enhanced passive surveillance. It is confusing because you are including surveillance information out of the “2.3 Surveillence data” section. I leave this to the discretion of the authors.
Line 178-179: Please correct the sentence, it is wrong and unclear. Is this the same methodology as in the mange areas? The number of days of the census in the mange areas is not described. I don’t understand the difference between this census and the one performed in the other mange areas, please clarify it better.
Line 210-211: This should refer to the minimum number of carcasses not detected by surveillance, as discussed before.
Line 217: Overall the use of terminology throughout the manuscript should be improved. The subheading “3.1. Description of the mange epidemic in the study area” is confusing, because Cavallazza area is also part of the study area, isn’t it? For consistency and to follow the aim of the study, wouldn’t be better to also include in the subheading the terms of ordinary or enhanced surveillance? I leave this to the discretion of the authors.
Line 256: The information from this paragraph and the next one in line 260 is still ordinary surveillance, it should be all presented in the same paragraph or clearly indicate in each one that you refer to the ordinary surveillance data. Besides, starting a paragraph with “However” is awkward.
Line 376: Please correct “young chamois are”.
Line 381: “very high” is as precise as “high”, which out of context is senseless and does not add any significance to the sentence. I suggest to just mention age-related mortality, but I leave this decision to the discretion of the authors.
Line 383: I understand the authors refers to “wild mountain ungulates”. To complement this discussion, there are studies that revealed that some specific pathogens can actually have a huge and higher impact in chamois population dynamics and population viability, see for example Serrano et al., 2015 Front. Microbiol. https://doi.org/10.3389/fmicb.2015.01307.
Line 399: I suggest deleting “too”.
Line 416: Why this sentence starts in bold letters?
Line 442: I encourage the authors to rewrite the conclusions of the study, for clarity but also to stretch the importance of the study results. For example, how environmental conditions affected surveillance, if the dynamics were similar to other studies or other ungulate species, variability of dynamics between mange areas, if mange was a significant cause of death, tendency to fade out, any evidence of geography affecting the dynamics of the mange outbreak, etc Some of these topics may deserve further focus in the discussion as well. I also found weird that the authors include in the conclusions a statement about the costs of the different types of surveillance considering that this information is not disclosed in the study, and it has barely been discussed in the article.
Table 1: According to Table S1 there are mange zones in which hunting is still banned. Did the epidemic wave finish in all the mange zones? If the epidemic wave was still going on in 2021 indicate it in the “Epidemic wave duration” column.
Figure 3: Please complete the caption, number of mange case detected in which type of surveillance?
Table S1: please review table for consistency in using capital letters or quotations marks. I do not understand some of the actions, what does it mean “As with no demographic increase” in the Management column? Why the Type 3 management has no aim? Type 7, what is an effective demography recovery? Does it mean that population size recovered to the levels before the epidemic? What % of the population is the regular hunting bag? The regular hunting bag needs to be mentioned in the text at least, but you could also include it in the table.
Author Response
Belluno, 06/08/2022
Dear Editor,
Please find enclosed the paper entitled “Description of a sarcoptic mange outbreak in Alpine chamois using an enhanced surveillance approach” by Federica Obber and co-authors, to be considered for publication in the Special Issue "Wildlife Disease Monitoring: Methods and Perspectives" of the journal Animals.
We are very pleased with the comments received and we think that observations from Reviewer 1 deserve to be addressed. We therefore include our response (in bold) to each specific comment below. Please find also attached the new revised manuscript, that we consider to be a significant improvement from the first one.
Sincerely
Federica Obber
Reviewer 1: Rebuttal letter (Round 2)
Comments and Suggestions for Authors – Reviewer 1 (Round 2)
The reviewed version of animals-1805990 has improved substantially, and the authors have addressed or justified satisfactorily most of my previous comments. However, I have further comments that needs to be addressed before recommending this manuscript to be published. The authors should make some further efforts to organize better and clarify the information.
For example, I do like the supplementary information they have provided, but the content in Table S1 is unclear.
We improved Table S1, as specified below
The authors did not disclose any information about the possibility of other ungulates being involved in the mange dynamics. Although I understand they want to focus on alpine chamois, in my opinion, they should state if other species were also detected with mange in the study area, and at least discuss if the Alpine ibex may have been involved in some of the dynamics described in chamois, for example, in those areas where sarcoptic mange was sporadically detected in chamois after few years of no detection.
The Alpine Ibex colonies of Pale di San Martino was very small, and nearly decimated by the mange epidemic wave affecting the chamois population. As such, this species most probably didn’t play any important role in the epidemic dynamics. A reintroduction project was carried out in the period 2000 - 2002 with the release of 30 subjects and the monitoring of colonies was implemented. Surveillance of sarcoptic mange and management activities for this protected species, in accordance with Italian legislation, has been done differently than chamois population and for this reason the article focused only on the chamois species.
The authors state in the conclusion that enhance passive surveillance is “too costly” to apply it in wide areas and for long periods, but the study does not provide any data about costs or labor hours of the two different types of surveillance, and it is not properly discussed in the text. In my opinion, providing some of this information, which should not be too difficult, would substantially improve the quality of the study as it is an important decision for managers. Nevertheless, the conclusions section is still poor, and I encourage the authors to rewrite it.
Modified as suggested
Specific comments:
Line 25: please correct “has been affected”
Done as suggested
Line 34-35: it may not be appropriate to include in the abstract a statement like this when the study does not assess costs of the different surveillance approaches, and it has barely been discussed in the study. In addition, I don’t know what the authors mean by an “active approach” in passive surveillance. I suggest rewriting these sentences, this is to the discretion of the authors, but the abstract does not stretch enough the content of the study.
Modified as suggested
Line 41: Please review the manuscript for double or missing spaces, there is a missing space after the point.
Corrected here and in lines: 47, 54, 99, 144
Line 49: “for its conservation”
Modified as suggested
Line 50: please review sentence for English grammar.
Changed to improve the clarity.
Line 52: “conservation” seems out of place in this sentence and it has been discussed in previous sentences, please correct.
Changed as: “…wildlife management, forcing stakeholders to adapt hunting bag size to cope with less abundant stocks.”
Line 53: I suggest adding a reference, there are several examples in the literature already cited.
We added a reference
Line 54: Does “Sarcoptic” need to be capitalized? Similar for line 58 and elsewhere.
We changed to lower case
Line 55: “influenced by the interaction”. Please review carefully the whole manuscript for English grammar.
Changed as “…and it interacts with…”
Line 56: is this statement supported by literature? It is a strong statement, the authors may want to phrase this sentence differently as mortality, in a longer temporal scale, also differ among naïve populations, as demonstrated by your own study.
Actually, in this introductory context, we wanted only to underline the different impact of sarcoptic mange in naïve and not naïve populations, given that it’s well demonstrated the difference in mortality rates between the two cases. For a matter of clarity, we changed as: “Life history of affected populations, namely previous contacts (or not) with the agent, also play an important role, as a first epidemic wave is likely to have remarkable effects on naïve populations.”
Line 58: “naïve populations” or “unaffected populations” have more sense to me than “areas”.
Changed to “naïve populations”
Line 77: I believe “summaries” is incorrect.
You are correct. Changed to “summarises” (also in Line 442)
Line 98: Does any of the reactive measures targeted the removal of diseased animals? For example, in the type 1 of hunting strategy? I think this is very important to clarify it as this has been a debate on how to manage mange outbreaks. Please state it clear.
No removal of diseases animals was applied in the area, but an increase of the hunting bag was allowed at the beginning of the epidemic wave, as a sort of anticipation of the forthcoming mortality. We believe that it’s not appropriate to develop further this aspect in the manuscript.
Line 99: consider capitalizing each sentence.
This paragraph is not intended to be an itemization, rather a descriptive list of actions. As such, every bullet point ended with a semicolon with the purpose not to break the flow, wich calls for lower case initials at the beginning.
Line 145: this sentence is unclear, please rewrite.
Changed as: “All populations of each mange zone were monitored in summer (July) by a single census day every other year. To account for years in which census was not carried out, the acquired dataset was filled in by calculating the arithmetic mean of the two adjacent data.”
Line 153: for what I understood the rangers did not perform a mange control, but rather a mange case detection/monitoring, etc please correct or explain what the mange control was.
Changed as “…mange monitoring only.”
Line 154: please delete “association of”.
Replaced with “both”, as removing it would result in an ambiguous sentence (and/or)
Line 168: “both study area and CA” Isn’t CA within the study area? This sentence is confusing, please correct.
It seemed better to specify that, while using two different surveillance protocols (ordinary/enhanced passive), the protocols for case report were the same. However, we agree that the sentence may arise confusion and therefore we removed the sentence “for both study area and CA”.
Line 177: “Cavallazza Area (CA)” is not a clear subheading, and it is confusing because you have already described this area. Please correct it to describe what information you are including in this section. The authors may also consider moving the census of CA to the demographic data section, and dedicate two separate sections to describe the ordinary passive surveillance and enhanced passive surveillance. It is confusing because you are including surveillance information out of the “2.3 Surveillence data” section. I leave this to the discretion of the authors.
We have changed the subheading (Line 177) as “2.4. Intensive census and enhanced surveillance in CA”. To improve clarity, we started the paragraph (Line 178) with “From 2008 to 2013, CA chamois population was investigated using an intensive census routine based on block count. Censuses were repeated every year, for five consecutive days, and were carried out by PPSN personnel…”. For consistency, we changed also the other subheading (Line 148) from “Surveillance data” to “Ordinary surveillance data”.
We left this paragraph 2.4 (Line 177) by itself, because if we moved the “intensive census part” under the demographic data section, we would also have to move the “enhanced surveillance part” under the surveillance session, and we think that this would turn out quite complicated.
Line 178-179: Please correct the sentence, it is wrong and unclear. Is this the same methodology as in the mange areas? The number of days of the census in the mange areas is not described. I don’t understand the difference between this census and the one performed in the other mange areas, please clarify it better.
The censing methodology in CA was the same as in mange areas, but it was more “intense” (e.g. every year instead of every other year; for five days instead of one). The number of census days for mange zones is specified at line 144. The sentence was rewritten as follows: “From 2008 to 2013, CA chamois population was investigated using block count. Censuses were repeated every year, for five consecutive days, and were carried out by PPSN personnel, Trento Forestry Department and gamekeepers along six transects (length 1986 ± 1760 m), encompassing the whole area to minimize the probability of underestimation [20]”
Line 210-211: This should refer to the minimum number of carcasses not detected by surveillance, as discussed before.
Modified as suggested
Line 217: Overall the use of terminology throughout the manuscript should be improved. The subheading “3.1. Description of the mange epidemic in the study area” is confusing, because Cavallazza area is also part of the study area, isn’t it? For consistency and to follow the aim of the study, wouldn’t be better to also include in the subheading the terms of ordinary or enhanced surveillance? I leave this to the discretion of the authors.
We agree with reviewer suggestion and changed the subheading at Line 217 as “3.1. Description of the mange epidemic using ordinary surveillance data”, and also changed subheading at Line 298 as “3.2. Description of mange epidemic in CA using enhanced surveillance data”
Line 256: The information from this paragraph and the next one in line 260 is still ordinary surveillance, it should be all presented in the same paragraph or clearly indicate in each one that you refer to the ordinary surveillance data. Besides, starting a paragraph with “However” is awkward.
We merged the paragraphs from 252 to 263, and changed 260 to “As is clear from Figure 4, the proportion…”
Line 376: Please correct “young chamois are”.
Done
Line 381: “very high” is as precise as “high”, which out of context is senseless and does not add any significance to the sentence. I suggest to just mention age-related mortality, but I leave this decision to the discretion of the authors.
Changed as: “…previous year, supporting the existence of a correlation between age and mortality”
Line 383: I understand the authors refers to “wild mountain ungulates”. To complement this discussion, there are studies that revealed that some specific pathogens can actually have a huge and higher impact in chamois population dynamics and population viability, see for example Serrano et al., 2015 Front. Microbiol. https://doi.org/10.3389/fmicb.2015.01307.
We added a new sentence and cited Serrano et al., 2015
Line 399: I suggest deleting “too”.
Done as suggested
Line 416: Why this sentence starts in bold letters?
Typo, corrected
Line 442: I encourage the authors to rewrite the conclusions of the study, for clarity but also to stretch the importance of the study results. For example, how environmental conditions affected surveillance, if the dynamics were similar to other studies or other ungulate species, variability of dynamics between mange areas, if mange was a significant cause of death, tendency to fade out, any evidence of geography affecting the dynamics of the mange outbreak, etc Some of these topics may deserve further focus in the discussion as well. I also found weird that the authors include in the conclusions a statement about the costs of the different types of surveillance considering that this information is not disclosed in the study, and it has barely been discussed in the article.
We introduced some new sentences in the Discussion section (Lines 421-427) to exemplify the differences in costs between ordinary and enhanced surveillance, and we briefly included the main outcome in the Conclusions. The latter section was further improved in other parts.
Table 1: According to Table S1 there are mange zones in which hunting is still banned. Did the epidemic wave finish in all the mange zones? If the epidemic wave was still going on in 2021 indicate it in the “Epidemic wave duration” column.
In some mange zones (PAN, LIT, STE, VFM) few mange cases were still reported in 2020. However, for some (PAN and LIT) these cases are a possible comeback or second wave of the mange epidemic (already commented at lines 329-331), whereas the few cases in STE and VFM probably represent the last cases at the end of the epidemic wave. The definition of the epidemic wave duration was improved in the first revision, and based on the stock recovery, which was already starting also in these areas, as demonstrated by the most recent census data (not shown in the manuscript, because referring to 2021). A specification has been added at the note 3 below the table to better clarify this point.
Figure 3: Please complete the caption, number of mange case detected in which type of surveillance?
Changed as “Trend in the number of mange cases (grayscale) detected through ordinary surveillance. The year of the mange index case (MIC) is represented in red.”
Table S1: please review table for consistency in using capital letters or quotations marks.
We have corrected typos
I do not understand some of the actions, what does it mean “As with no demographic increase” in the Management column?
You made me spot an error in the table, and therefore replaced “increase” with “decrease”; I think it is now clear that management would be carried out as if chamois population remained numerically stable
Why the Type 3 management has no aim?
The Mange Strategy cites the local (Trento Province) Hunting Law, in particular a “general” Article that says how hunting should promote the environment, including no specific indications; I changed as “No hunting restrictions”
Type 7, what is an effective demography recovery? Does it mean that population size recovered to the levels before the epidemic?
We changed Type7 row as follows:
PRIORITARY MANAGEMENT AIMS
Promote demographic recovery (≥50% of pre-epidemic status).
MANAGEMENT
Hunting must return to normality following these steps:
- 50-70% of pre-epidemic stocks: hunting bag ≤5% of census data
- 70-80% of pre-epidemic stocks: hunting bag ≤10% of census data
- ≥80%: ordinary hunting bag
What % of the population is the regular hunting bag? The regular hunting bag needs to be mentioned in the text at least, but you could also include it in the table.
We included the information in the Study area section, changing the paragraph from line 94 as follows: “Chamois is a renowned game species overall Trento province; for hunting purposes, populations are monitored at local level using block count; the data is then aggregated by larger areas, inside which the overall hunting pressure must not exceed 15% of the census population, and usually settles at about 10%. Management of chamois population underwent significant change in the study area, since the detection of the first mange cases in 2005: the study area has been further subdivided…”.
